# Knowledge Graph Simple Question Answering
# for Unseen Domains

**Georgios Sidiropoulos**                    G.SIDIROPOULOS@UVA.NL
**Nikos Voskarides**                    NICKVOSK@GMAIL.COM
**Evangelos Kanoulas**                    E.KANOULAS@UVA.NL
*University of Amsterdam, The Netherlands*

## Abstract

Knowledge graph simple question answering (KGSQA), in its standard form, does not take into account that human-curated question answering training data only cover a small subset of the relations that exist in a Knowledge Graph (KG), or even worse, that new domains covering unseen and rather different to existing domains relations are added to the KG. In this work, we study KGSQA in a previously unstudied setting where new, unseen domains are added during test time. In this setting, question-answer pairs of the new domain do not appear during training, thus making the task more challenging. We propose a data-centric domain adaptation framework that consists of a KGSQA system that is applicable to new domains, and a sequence to sequence question generation method that automatically generates question-answer pairs for the new domain. Since the effectiveness of question generation for KGSQA can be restricted by the limited lexical variety of the generated questions, we use distant supervision to extract a set of keywords that express each relation of the unseen domain and incorporate those in the question generation method. Experimental results demonstrate that our framework significantly improves over zero-shot baselines and is robust across domains.

## 1. Introduction

Large-scale structured Knowledge Graphs (KGs) such as Freebase [Bollacker et al., 2008] and Wikidata [Pellissier Tanon et al., 2016] store real-world facts in the form of subject–relation–object triples. KGs are being increasingly used in a variety of tasks that aim to improve user experience [Bota et al., 2016]. One of the most prominent tasks is Knowledge Graph Simple Question Answering (KGSQA), which aims to answer natural language questions by retrieving KG facts [Yih et al., 2015]. In practice, many questions can be interpreted by a single fact in the KG. This has motivated the KGSQA task [Bordes et al., 2015, Mohammed et al., 2018, Petrochuk and Zettlemoyer, 2018], which is the focus of this paper. In KGSQA, given a *simple* question, e.g. "who directed the godfather?", the system should interpret the question and arrive at a single KG fact that answers it: (The Godfather (film), film.film.directed_by, Francis Ford Coppola).

KGSQA systems are trained on manually annotated datasets that consist of question-fact pairs. In practice, the applicability of such systems in the real-world is limited by two factors: (i) modern KGs store millions of facts that cover thousands of different relations, but KGSQA training datasets can only cover a small subset of the existing relations in the KG  [ElSahar et al., 2018], and (ii) KGs are dynamic, i.e. they are updated with new domains that cover new relations [Pellissier Tanon et al., 2016]. Solving (i) and (ii)

by exhaustively gathering question-fact pair annotations would be prohibitively laborious, thereby we need to rely on automatic methods.

Motivated by the above, in this work, we study the KGSQA task in a setting where we are interested in answering questions about a new, unseen domain that covers relations, for which we *have* instances in the KG, but we *have not seen* any question-fact pair during training. We model this as a domain adaptation task [Mansour et al., 2009, Pan and Yang, 2010] and propose a data-centric domain adaptation framework to address it. Data-centric domain adaptation approaches focus on transforming or augmenting the training data, instead of designing specialized architectures and training objectives as model-centric domain adaptation approaches do [Chu and Wang, 2018]. Our framework consists of: (a) a KGSQA system which can handle the unseen domain, and (b) a novel method that generates training data for the unseen test domain.

The KGSQA system we introduce performs mention detection, entity candidate generation and relation prediction on the question, and finally selects the fact that answers the question from the KG. To improve relation prediction on questions that cover relations of the unseen domain, we automatically generate synthetic questions from KG facts of the unseen domain (i.e. knowledge graph question generation – QG). The resulting synthetic question-fact pairs are used to train the KGSQA system for the unseen domain. We find that the effectiveness of QG for KGSQA can be restricted not only by the quality of the generated questions, but also by the lexical variety of the questions. This is because users ask questions underlying the same relation using different lexicalizations (e.g. "who is the author of X", "who wrote X"). To address this, we use distant supervision to extract a set of keywords for each relation of the unseen domain and incorporate those in the question generation method.

Our main contributions are the following: (i) we introduce a new setting for the KGSQA task, over new, previously unseen domains, (ii) we propose a data-centric domain adaptation framework for KGSQA that is applicable to unseen domains, and (iii) we use distant supervision to extract a set of keywords that express each relation of the unseen domain and incorporate them in QG to generate questions with a larger variety of relation lexicalizations. We experimentally evaluate our proposed method on a large-scale KGSQA dataset that we adjust for this task and show that our proposed method consistently improves performance over zero-shot baselines and is robust across domains.[1]

## 2. Problem Statement

Let $E$ denote the set of entities and $R$ the set of relations. A KG $K$ is a set of facts $(e_s, r, e_o)$, where $e_s, e_o \in E$ are the subject and object entities respectively, and $r \in R$ is the relation between them. Each relation $r$ has a unique textual label $r_l$ and falls under a single domain $\mathcal{D}$. For instance, music.album.release_type and music.artist.genre fall under the Music domain. *Simple* questions mention a single entity and express a single relation. For instance, the question "who directed the godfather?" mentions the entity "The Godfather" and expresses the relation film.film.directed_by.

Given a *simple* question $q$ that consists of a sequence of tokens $t_1, t_2, \ldots, t_T$, the KGSQA task is to retrieve a fact $(\hat{e}_s, \hat{r}, \hat{e}_o)$, where $(\hat{e}_s, \hat{r})$ accurately interprets $q$ (i.e., $\hat{e}_s$ is mentioned

---

1. Our code is available at https://github.com/GSidiropoulos/kgsqa_for_unseen_domains.

in $q$ and $\hat{r}$ is expressed in $q$) while $\hat{e}_o$ provides the answer to $q$. In our setting, we aim to build a KGSQA system that can perform well on a previously unseen domain. A domain is "unseen" when facts that cover relations of that domain do exist in $K$, but gold-standard question-fact pairs of that domain do *not* appear in the training data. This setting is an instance of domain adaptation, where a model is trained on data $\mathcal{S}$, which is drawn according a source distribution, and tested on data $\mathcal{T}$ coming from a different target distribution. Domain adaptation over KG domains is more challenging compared to domain adaptation over single KG relations [Yu et al., 2017, Wu et al., 2019], because it is less likely for relations with similar lexicalizations to appear in the training set.

## 3. KGSQA system

In this section, we detail our KGSQA system. In Section 4, we will describe how we generate synthetic training data to make this system applicable to unseen domains. Following current state-of-the-art on KGSQA [Petrochuk and Zettlemoyer, 2018], we split the task into four sub-tasks, namely, *entity mention detection* (MD), *entity candidate generation* (CG), *relation prediction* (RP), and *answer selection* (AS). The skeleton of our KGSQA system generally follows previous work, and we modify the MD and RP architectures.

**Mention Detection (MD)** Given the question $q$, MD outputs a single entity mention $m$ in $q$, where $m$ is a sub-sequence of tokens in $q$. We model this problem as sequence tagging, where given a sequence of tokens, the task is to assign an output class for each token [Huang et al., 2015, Lample et al., 2016]. In our case, the output classes are entity (E) and context (C). For instance, the correct output for the question "who directed the godfather?" is "[C C E E]". We use a BiLSTM with residual connections (R-BiLSTM) [He et al., 2016], since it outperformed vanilla RNN, BiRNN, and a CRF on top of a BiRNN [Petrochuk and Zettlemoyer, 2018] in preliminary experiments.

**Candidate Generation (CG)** Given the mention $m$ extracted from the previous step, CG maps $m$ to a set of candidate entities $C_S \subset E$. For instance, CG maps the mention "the godfather" to the entities { The Godfather(film), The Godfather(book)... }. The CG method we use was proposed in Türe and Jojic [2017]. Briefly, the method pre-builds an inverted index $I$ from n-grams of mentions to entities, and it looks-up the n-grams of $m$ in $I$ to obtain $C_S$.

**Relation Prediction (RP)** Given the question $q$ and the set of entities $C_s$ extracted in the previous step, RP outputs a single relation $\hat{r} \in R$ that is expressed in $q$. Previous work models RP as a large-scale multi-label classification task where the set of output classes is fixed [Petrochuk and Zettlemoyer, 2018]. In our domain adaptation scenario, however, we want to be able to predict relations that we have not seen during training. Therefore, we model RP as a relation ranking task, as in [Yu et al., 2017], and use the textual label $r_l$ to represent the relation $r$ (instead of using a categorical variable). This way we can in principle represent any relation $r \in R$ during inference time. Below we describe the architecture we use for RP and how we perform training and inference. Our architecture is a simpler version of [Yu et al., 2017], where they model a relation both as a sequence and a categorical variable, and they use more complex sequence encoders.

First we describe how we encode the question $q$ and the relation $r$. In order to generalize beyond specific entity names, we first replace the previously detected entity mention $m$ in $q$ with a placeholder token; e.g "who directed SBJ". We then map each term to its embedding and feed the word embeddings to an LSTM; embeddings are initialized with pretrained word2vec embeddings [Mikolov et al., 2013]. The final hidden state of the LSTM $\boldsymbol{\gamma}^{(q)}$ is used as the encoding of the question. In order to represent $r$, we use its label $r_l$ (e.g. film.film.directed_by). Similarly with the question encoding, we encode $r_l$ with an LSTM to obtain $\boldsymbol{\gamma}^{(r)}$. However, since questions and relations significantly differ both grammatically and syntactically, the two LSTM encoders do not share any parameters. The ranking function $f$ is calculated as $f(q,r) = \cos(\boldsymbol{\gamma}^{(q)}, \boldsymbol{\gamma}^{(r)})$, where $\cos(\cdot)$ is the cosine similarity.

We train $f$ using standard pairwise learning to rank. The loss is defined as follows:

$$L(\theta) = \sum_{r} \sum_{r' \in R'} \max(0, \mu - f(q,r) + f(q,r')), \tag{1}$$

where $\theta$ are the parameters of the model, $\mu$ is a hyperparameter, and $R'$ is the set of sampled negative relations for a question $q$. We design a specialized *negative sampling* method to select $R'$. With probability $P_R^-$ we uniformly draw a sample from $R^- = \{r'|r' \in R \wedge r' \neq r\}$; the set of all available relations except the positive relation $r$. With probability $1 - P_R^-$ we draw a random sample from $\hat{R}^- = \{r'|r' \in \mathcal{D}_R^+ \wedge r' \neq r\}$; the set of relations that are in the same domain as the positive relation $r$. This way, we expose the model to conditions it will encounter during inference. At inference time, given a question $q$ and a set of relations we score all question-relation pairs $(q,r)$ with $f$ and select the relation $\hat{r}$ with the highest score. Unfortunately, computing a score with respect to all possible relations in $R$ leads to poor performance when there is no linguistic signal to disambiguate the choice. In order to address this issue, we constrain the set the potential output relations $R_c$ to be the union of the relations expressed in the facts where the entities in $C_S$ participate in [Petrochuk and Zettlemoyer, 2018]. Formally, we define the target relation classes to be $R_c = \{r \in R|(e_s, r, e_o) \in K \wedge e_s \in C_S\}$. For example, given the question "who directed the godfather", the potential relations are { film.film.directed_by, book.written_work.author, ... }. Using the aforementioned constraint we can safely ignore relations like tv.tv_series_episode.director by taking into account that The Godfather does not appear in any tv-related facts.

**Answer Selection (AS)** Given the set of entities $C_S$ obtained from CG, and the top ranked relation $\hat{r}$ obtained from RP, AS selects a single fact $(\hat{e}_s, \hat{r}, \hat{e}_o)$, where $\hat{e}_o$ answers the question $q$. The set of candidate answers may contain more than one facts $(e'_s, \hat{r}, e'_o)$, where $\forall e'_s \in C_S$. Since there is no explicit signal on which we can rely to disambiguate the choice of subject, all the potential answers are equally probable. We therefore use a heuristic based on popularity, introduced by Mohammed et al. [2018]: we choose $\hat{e}_s$ to be the entity that appears the most in the facts in $K$ either as a subject or as an object. Having $\hat{e}_s$ and $\hat{r}$ we can retrieve the fact $(\hat{e}_s, \hat{r}, \hat{e}_o)$. For our running example ("who directed the godfather"), given film.film.directed_by (from RP) and entities { The Godfather(film), The Godfather(book)... } (from CG) we can select the fact (The Godfather (film), film.film.directed_by, Francis Ford Coppola).

## 4. KGSQA to unseen domains using question generation

Even though all the components of the aforementioned KGSQA system were designed to work with unseen domains, preliminary experiments demonstrated that RP does not generalize well to questions originating from unseen domains. This is expected since RP is a large-scale problem (thousands of relations), and it is very challenging to model less frequent or even unknown relations that are expressed with new lexicalizations.

We therefore focus on improving RP for questions originating from unseen domains. Inspired from the recent success of data-centric domain adaptation in neural machine translation [Chu and Wang, 2018], we perform synthetic question generation from KG facts of the unseen domain to generate question-fact pairs for training the RP component (see Section 3).[2] In the remainder of this section we briefly describe the base question generation (QG) model we build upon and how we augment the model to more effectively use textual evidence and thus better generalize to relations of the unseen domain.

### 4.1 Base model for QG

Given a fact $(e_s, r, e_o)$ from the target domain, QG aims to generate a synthetic question $\hat{q}$. During training, only question-fact pairs from the known domains are used. Our base model is the state-of-the-art encoder-decoder architecture for QG [ElSahar et al., 2018]. It takes as input the fact $(e_s, r, e_o)$ alongside with a set of textual contexts $C = \{c_s, c_r, c_o\}$ on the fact. Those textual contexts are obtained as follows: $c_s$ and $c_o$ are the types of entities $e_s$ and $e_o$ respectively, whereas $c_r$ is a lexicalization of the relation $r$ obtained by simple pattern mining on Wikipedia sentences that contain instances of $r$. For instance, given the fact (The Queen Is Dead, music.album.genre, Alternative Rock), the textual contexts are: $c_s = \{\text{"album"}\}$, $c_r = \{\text{"album by"}\}$ and $c_o = \{\text{"genre"}\}$.

The encoder maps $e_s$, $r$ and $e_o$ to randomly initialized embeddings and concatenates those to encode the whole fact. Also, it encodes the text in $c_s$, $c_r$ and $c_o$ separately using RNN encoders. The decoder is a separate RNN that takes the representation of the fact and the RNN hidden states of the textual contexts to generate the output question $\hat{q}$. It relies on two attention modules: one over the encoded fact and one over the encoded textual contexts. The decoder generates tokens not only from the output vocabulary but also from the input (using a copy mechanism) to deal with unseen input tokens.

### 4.2 Using Richer Textual Contexts for QG

The role of the textual contexts $C$ in the aforementioned base model is critical, since it enables the model to provide new words/phrases that would have been unknown to the model otherwise [ElSahar et al., 2018]. Even though the base model generally generates high quality questions, in our task (KGSQA), we aim to generate a larger range of lexicalizations for a single relation during training in order to generalize better at test time. This is because users with the same intent may phrase their questions using different lexicalizations (e.g. "who is the author of X", "who wrote X"). Thus, in this section we focus on how to provide

---

2. Note that Dong et al. [2017] also performed QG for improving the overall KGSQA performance. However, their model is not applicable to our domain adaptation scenario since their model relies on modifying existing questions and all domains were predefined.

**Table 1:** Examples of relation textual contexts extracted by our keyword extraction approach.

| Relation | Textual Context |
|---|---|
| music.artist.label | records, artists, album, released, label, signed, band |
| film.film.directed_by | film, director, directed, films, short, directing, producer |
| people.deceased_person.place_of_death | died, death, deaths, born, age, people, male, actors |

the model with a diverse set of lexicalizations for a relation $r$ instead of a single one as in the base model, in order to be able to generate a more diverse set of questions in terms of relation lexicalizations. More precisely, given a relation $r$, we extract $k$ keywords that will constitute the relation's textual context $c_r$. To this end, we first extract a set of candidate sentences $S_r$ that express a specific relation $r$ between different pairs of entities. Second, we extract keywords from the set $S_r$, rank them and select the top-$k$ keywords that constitute the set $c_r$. We detail each of these steps below.

**Extracting sentences** Given a set of facts $F_r$ of relation $r$ between different pairs of entities, we aim to extract a set of sentences $S_r$, where each sentence $s \in S_r$ expresses a single fact $(e_s, r, e_o)$ in $F_r$ [Voskarides et al., 2015]. For this, for each fact $(e_s, r, e_o)$ in $F_r$, we need to (a) extract a set of candidate sentences $S$ that might express $(e_s, r, e_o)$ and (b) select the sentence that best expresses the relation. For (a), we collect the set of sentences $S$ using distant supervision, similarly to [Mintz et al., 2009]: $S$ consists of sentences that mention $e_o$ in the Wikipedia article of $e_s$ and sentences that mention $e_s$ in the Wikipedia article of $e_o$. For (b), we score each sentence $s \in S$ w.r.t. the label $r_l$ of the relation $r$ using the cosine similarity $cos(e(s), e(r_l))$, where $cos(\cdot)$ is the cosine similarity and $e(x)$ is calculated as $e(x) = (1/|x|) \sum_{t \in x} w_t$, where $w_t$ is the embedding of word $t$. Finally, we take the sentence $s'$ with the highest score and add it to the set $S_r$.

**Extracting keywords** After extracting the set of sentences $S_r$, we aim to extract the set of keywords $c_r$. For this, we treat $S_r$ as a single document and score each word $t$ that appears in $S_r$ using tf-idf, $score(t) = tf(t, S_r) \cdot idf(t, S_R)$, where $S_R$ is the union of all $S_{r'}, r' \in R$. The top-$k$ scoring words constitute the set of keywords $c_r$. Table 1 depicts example keywords generated by the procedure described above.

The keyword extraction approach described above is conceptually simple yet we later show that it significantly improves upon the base model when applied to KGSQA.

## 5. Experimental Setup

In this section, we discuss how we design the experiments to answer the following research questions: **RQ1**) How does our method for generating synthetic training data for the unseen domain perform on RP compared to a set of baseline methods? **RQ2**) How does our full method perform on KGSQA for unseen domains compared to state-of-the-art zero-shot data-centric methods? **RQ3**) How does our data-centric domain adaptation method compare to a state-of-the-art model-centric method on RP?

**Dataset** In our experiments we use the SimpleQuestions dataset, which is an established benchmark for studying KGSQA [Bordes et al., 2015]. The dataset consists of 108,442

questions written in natural language by human annotators, paired with the ground truth fact that answers the question. The ground truth facts originate from Freebase [Bollacker et al., 2008]. The dataset covers 89,066 unique entities, 1,837 unique relations and 82 unique domains. In our setup, we leave one domain out to simulate a new, previously unseen domain, and train on the rest. We choose six challenging domains as target domains: Film, Book, Location, Astronomy, Education and Fictional Universe; the first three are among the largest domains and the last three are medium-sized. The aforementioned domains are challenging because they have very low overlap in terms of relation lexicalization w.r.t. the rest of the domains used as source domains. The training data consists of the question-fact pairs that appear in the source domains, augmented with synthetically generated data of the target/unseen domain. In practice, we replace all questions from the target domain that initially appear in the full training set with their corresponding synthetically generated questions.[3] As a source of text documents for the textual context collection (see Section 4.2), we use Wikipedia articles augmented with dense entity links provided by DAWT [Spasojevic et al., 2017].

**Baselines**  To answer **RQ1**, we keep the KGSQA system unchanged and alternate the way of generating synthetic questions. We compare the RP performance on the unseen domain given the following ways of generating synthetic data of the unseen domain: (i) No synthetic data, (ii) Wiki-raw-sentences: uses the raw Wikipedia sentence that expresses the ground truth fact that answers the question (automatically extracted using the procedure in Section 4.2), and (iii) the state-of-the-art QG method proposed in ElSahar et al. [2018]. To answer **RQ2**, we replace our RP component with two state-of-the-art RP models: (i) Petrochuk and Zettlemoyer [2018], which uses a BiLSTM to classify relations, and (ii) Yu et al. [2017], a zero-shot RP model that uses a HR-BiLSTM and is specifically designed to deal with unseen or less frequently seen relations. To answer **RQ3**, we compare the performance of our data-centric model on RP against a state-of-the-art model-centric zero-shot approach [Wu et al., 2019]: it the HR-BiLSTM proposed by Yu et al. [2017] with an adversarial adapter combined and a reconstruction loss. The adapter uses embeddings trained on Freebase and Wikipedia by JointNRE [Han et al., 2018] and learns representations for both seen and unseen relations.

**Evaluation metrics**  We run the experiments three times and report the median (only marginal and not significant differences were found among different runs) [Mohammed et al., 2018]. In contrast to the classic KGSQA where the task is to retrieve a single entity, it is standard practice when using the SimpleQuestions dataset to treat the problem as question interpretation [Petrochuk and Zettlemoyer, 2018]. More specifically, the objective is to rewrite the natural language question in the form of subject-relation pair. We evaluate our overall approach in terms of top-1 accuracy, i.e. whether the retrieved subject-relation pair matches the ground truth. We measure accuracy both at a macro- (domains) and at a micro-level (samples). Statistical significance is determined using a paired two-sided t-test.

---

3. One may hypothesize that since entities can appear in multiple domains (e.g. actors who are also singers), question generation becomes an unrealistically simple task. However, this is not the case because in our dataset, the entity overlap between seen and unseen domains is *only* 4.6%.

**Table 2:** Relation Prediction accuracy w.r.t. different ways of generating synthetic training data for the unseen domain. ▲ indicates a significant increase in performance compared to the top performing baseline ($p < 0.01$).

| Synthetic training data | Macro-avg. Accuracy (%) | Micro-avg. Accuracy (%) |
|---|---|---|
| - | 30.21 | 29.06 |
| Wiki-raw-sentences | 37.89 | 36.51 |
| QG [ElSahar et al., 2018] | 67.52 | 69.78 |
| QG (Ours) | **69.86▲** | **70.95 ▲** |

**Parameter configurations**  We initialize word embeddings with pretrained Google News 300-dimensional embeddings [Mikolov et al., 2013]. We use the Adam optimizer [Kingma and Ba, 2014]. Our MD model consists of 2 hidden layers, 600 hidden units, 0.4 dropout rate, frozen embeddings, and learning rate of $10^{-3}$; 50 training epochs. For the RP model, we use 1 layer encoder for both questions and relations that consists of 400 hidden units, with a frozen embedding layer, and a learning rate of $10^{-3}$; trained for 10 epochs. During training, we sample 10 negative questions per question using the procedure described in Section 3. We use a batch size of 300 and 200 for MD and RP respectively. For the QG model [ElSahar et al., 2018] and the model-centric RP [Wu et al., 2019] model we compare against, we use the hyperparameters as presented in their work. Note that for both our method and the baselines, the hyperparameters were tuned on the initial split of the SimpleQuestions dataset. We keep the parameters fixed for both our method and the baselines for all source-target domain setups. We set the number of keywords for each relation $k = 10$ (Section 4.2).

## 6. Results and Discussion

In this section we present and discuss our experimental results. All models under comparison have all their components fixed, except RP. Therefore, any improvement observed, is due to RP.

**Effect of synthetic data on RP (RQ1)**  Here we compare the RP performance of our method for generating synthetic training data with a set of baselines. For this experiment, the RP component of the KGSQA system remains unchanged and we only alter the data it is trained with. Table 2 shows the results. We observe that our QG method is the best performing one. It significantly outperforms the baseline QG method, which confirms that our method for generating rich textual contexts for relations (Section 4.2) is beneficial for KGSQA. As expected, Wiki-raw-sentences performs better than when not using training data from the target domain at all but performs much worse than the QG methods. This is expected since Wikipedia sentences are very different both syntactically and grammatically from the real questions that the KGSQA system encounters during test time. Next, we investigate how RP performance varies depending on the number of the target domain questions used to augment the training set. Figure 1 shows the results. First, we observe that, in the low data regime (less than 100 questions), the gap in performance between training with gold or synthetic questions is small. This is encouraging for applying our

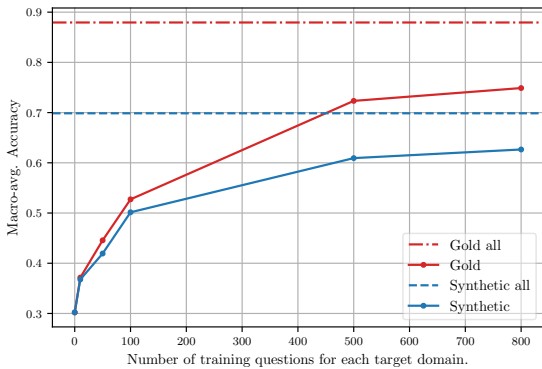

**Figure 1:** RP macro-accuracy when varying the number of target domain questions used to augment the training set. Gold refers to the gold standard questions and synthetic refers to the automatically generated questions. Gold all (Synthetic all) refers to the full set of training gold (synthetic) questions. The training set size of the smallest domain is 800, thus we report performance up to that point.

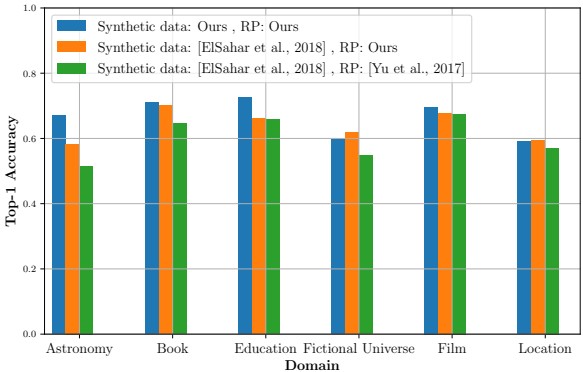

**Figure 2:** End-to-end accuracy on the KGSQA task per domain.

framework on domains in the long tail. As the number of questions increases, the performance for both gold and synthetic increase, however the gap between them increases, which is expected.

**Overall KGSQA performance for data-centric methods (RQ2)**  Next, we compare our full framework to variations that use state-of-the-art RP models. Table 3 shows the results. We observe that our full method (second to last row) improves over all the baselines and significantly outperforms the best performing baseline. As expected, we see that even though our full method holds strong generalization ability for unseen domains, there is a gap in the performance when using the automatically generated synthetic questions (second to last row) or the human generated questions (last row). This gap suggests that there is room for improvement for QG. Next, we test the systems under comparison in terms of generalization ability across domains. Figure 2 shows the results. First, we observe that our method achieves an accuracy of at least 60% for all domains which shows that it is robust across domains. Also, it outperforms the baselines in all but one domain. In order to gain further insights, we sampled success and failure cases from the test set. We found that the errors in the failure cases generally originate from the fact that the model relies

**Table 3:** End-to-end accuracy on the KGSQA task. ▲ indicates a significant increase in performance compared to the top performing baseline ($p < 0.01$). † is for ElSahar et al. [2018], ‡ for Petrochuk and Zettlemoyer [2018], and ⋆ for Yu et al. [2017].

| Synthetic training data | Relation Prediction (RP) | Macro-avg. Accuracy (%) | Micro-avg. Accuracy (%) |
|---|---|---|---|
| QG † | BiLSTM ‡ | 55.49 | 55.11 |
| | HR-BiLSTM ⋆ | 60.20 | 62.77 |
| | Ours | 63.90 | 65.18 |
| QG (Ours) | Ours | **66.49**▲ | **66.64**▲ |
| Gold Questions | Ours | 84.56 | 82.87 |

on lexicalizations that are frequent in the seen domains. We show such cases in Table 4. Furthermore, our analysis showed that one way of improving QG is to improve keyword extraction by collecting a larger set of relevant sentences that express a single relation, possibly by looking into other sources of text (e.g. news articles). In Table 5, the automatic evaluation results for the synthetic questions generated by our QG model against those generated by [ElSahar et al., 2018] further strengthen our claim. As can be seen from the table, our model outperforms the baseline, indicating that using a larger set of relevant sentences for a single relation, can be beneficial to the generated questions.

**Table 4:** Examples of success cases (top 3 rows) and failure cases (bottom 3 rows) of our QG method.

| Unseen Domain | Gold Questions | Synthetic Questions |
|---|---|---|
| Astronomy | what is something that carolyn shoemaker discovered | what is the astronomical objects discovered by carolyn shoemaker |
| Book | what's the subject of the cognitive brain | what is the subjects of the written work the cognitive brain |
| Location | which country is cumberland lake located in | where is cumberland lake located |
| Film | in what country did the film joy division take place | what country is joy division under |
| Book | who authored the book honor thyself | who was the director of the book honor thyself |
| Location | what was a historic atlantic city convention hall team | what event took place at historic atlantic city convention hall in 1943 |

**Comparison to a model-centric method (RQ3)** Here, we compare our data-centric method for domain adaptation to a state-of-the-art model-centric method on RP [Wu et al., 2019]. In order to perform a fair comparison when testing for RP, we follow their setup (see Section 5.1. in [Wu et al., 2019]) and for this particular experiment we assume that MD and CG produce the correct output. Our method outperforms their method both on macro-accuracy (75.54% vs 75.02%), and micro-accuracy (77.08% vs 72.17%). Note that we use randomly initialized embeddings whereas in their work they use JointNRE relation embeddings trained on Wikipedia and Freebase, which provides an advantage to

**Table 5:** Automatic evaluation of question generation w.r.t. BLEU.

|  | BLEU-1 | BLEU-2 | BLEU-3 | BLEU-4 |
|---|---|---|---|---|
| QG ([ElSahar et al., 2018]) | 44.04 | 28.63 | 16.50 | 9.13 |
| QG (Ours) | **44.27** | **29.55** | **17.73** | **10.16** |

their method. Also note that their method (model-centric) is orthogonal to ours (data-centric) and therefore, an interesting future work direction would be to explore how to combine the two methods to further improve performance.

**Qualitative error analysis**   In order to gain insights on how each part of the pipeline affects the final prediction, we perform an empirical error analysis. We sample 60 examples for which our system provided a wrong answer (10 for each target domain), and investigate what led to the wrong prediction. Out of these examples, 43 mistakes were due to RP, 6 due to ED, 5 due to CG and 6 due to AS. For RP, 22 were assigned to conceptually similar relations within the target domain and 15 to similar relations outside the target domain, while the rest were assigned to a common relation outside the target domain. For ED, 4 were due to predicting an extra token as part of the entity mention and 2 were due to missing a token from the entity mention. For CG, 2 were because the gold entity was not part of the mention candidates and 1 was due to an error in the human annotation; the early termination proposed in [Türe and Jojic, 2017], is responsible for the rest. From this analysis, we confirm that RP remains the most challenging part of KGSQA. Within RP, what seems to be the challenge is that there are relations for which there is a high lexical similarity between the corresponding questions, but also between the relations per se.

**KGSQA performance on seen domains**   Finally, even though the focus of this paper is to perform KGSQA on unseen domains and thus we do not aim to improve state-of-the-art on seen domains, we also test our KGSQA system on the standard split of the SimpleQuestions dataset. Our model achieves a top-1 accuracy of 77.0%, which is ranked third among the state-of-the-art methods while having a simpler method than the two top-performing ones ([Petrochuk and Zettlemoyer, 2018]), [Gupta et al., 2018]). We provide a thorough comparison w.r.t. the state-of-the-art on seen domains in Table 6.[4]

## 7. Related Work

Methods on the standard KGSQA task are split to those following a pipeline approach–MD, CG, RP & AS– ([Türe and Jojic, 2017, Mohammed et al., 2018, Petrochuk and Zettlemoyer, 2018]) or an end-to-end approach [Lukovnikov et al., 2017, Gupta et al., 2018]. In our work we follow the former approach for solving KGSQA on unseen domains, since we found that all the components except RP are relatively robust for unseen domains. We leave the exploration of end-to-end approaches for our task for future work. More related our setting, Yu et al. [2017] and  Wu et al. [2019] tackle RP for KGSQA on unseen relations

---

4. Note that Zhao et al. [2019] reported an accuracy of 85.44%. However, they calculate accuracy w.r.t. the correctness of the object entity, which is not standard when testing on the SimpleQuestions dataset (see Section 5). When we calculate accuracy that way,  [Petrochuk and Zettlemoyer, 2018] achieves an accuracy of 91.50% and our method achieves 87.31%.

**Table 6:** Top-1 KGSQA accuracy on seen domains.

| Model | Accuracy (%) |
|---|---|
| Random guess ([Bordes et al., 2015]) | 4.9 |
| Memory NN ([Bordes et al., 2015]) | 62.7 |
| Attn. LSTM ( [He and Golub, 2016]) | 70.9 |
| GRU ([Lukovnikov et al., 2017]) | 71.2 |
| BiGRU-CRF & BiGRU [Mohammed et al., 2018] | 74.9 |
| CNN & Attn. CNN & BiLSTM-CRF [Yin et al., 2016] | 76.4 |
| HR-BiLSTM & CNN & BiLSTM-CRF [Yu et al., 2017] | 77.0 |
| **Ours** | **77.0** |
| BiLSTM-CRF & BiLSTM [Petrochuk and Zettlemoyer, 2018]) | 78.1 |
| Solr & TSHCNN [Gupta et al., 2018] | 80.0 |

(instead of whole domains). Both are model-centric domain adaptation approaches, while ours is data-centric. We experimentally showed that we outperform both in the setting of KGSQA on unseen domains. An interesting direction for future work would be to combine model-centric and data-centric approaches for our task.

More broadly, our work is also related to cross-domain semantic parsing [Su and Yan, 2017, Yu et al., 2018, Herzig and Berant, 2018, Zhang et al., 2019]. In contrast to the aforementioned line of work that maps questions to executable logical forms, we focus on questions that can be answered with a single KG fact.

## 8. Conclusion

In this paper, we proposed a data-centric domain adaptation framework for KGSQA that is applicable to unseen domains. Our framework performs QG to automatically generate synthetic training data for the unseen domains. We propose a keyword extraction method that when integrated in our QG model, it allows it to generate questions of various lexicalizations for the same underlying relation, thus better resembling the variety of real user questions. Our experimental results on the SimpleQuestions dataset show that our proposed framework significantly outperforms state-of-the-art zero-shot baselines, and is robust across different domains. We found that there is room for further improving QG particularly for KGSQA, which is a promising direction for future work.

## Acknowledgements

This research was supported by the NWO Innovational Research Incentives Scheme Vidi (016.Vidi.189.039), the NWO Smart Culture - Big Data / Digital Humanities (314-99-301), NWO under project nr CI-14-25, the H2020-EU.3.4. - Societal Challenges - Smart, Green And Integrated Transport (814961), the Google Faculty Research Awards program. All content represents the opinion of the authors, which is not necessarily shared or endorsed by their respective employers and/or sponsors.

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
