# OpenReview forum: "Knowledge Graph Simple Question Answering for Unseen Domains"
_AKBC.ws/2020/Conference — AKBC 2020_

### Official Review · AnonReviewer1 · 2020-03-26
**Well written and motivated paper. Simple and clear approach. Convincing results. Presentation, especially of experiments, should be improved.**

**Rating:** 8
**Confidence:** 4

**Review:**

The paper studies the problem of single-relation QA. It proposes a data generation technique for domain adaptation. The model is built using a known decomposition of the problem, with emphasis on making relation prediction easy to re-train using generated data. Additional question data is generated to re-train the model for previously unseen domains.

Overall, the paper is well written and motivated. The results are convincing. The paper is narrow and focused in its contribution, but the problem is significant enough to merit such focused contribution.

There are some issues it would be good to get the authors input on:

(a) The paper provides few examples and no figures. Both would have done much to illustrate the approach and make the paper more accessible. In fact, except the intro, there is not a single question example. This is also critical in the experimental results, which provide little insight into the numerical results (e.g., through qualitative analysis).

(b) There's no evaluation (or even examples) of the generated questions. Ideally, this should be done with human evaluation. This can really help understand the gap between the system performance and oracle questions.

(c) During experiments, do you train once and then tune and test for each unseen domain? Or do you include the other 5 unseen domains in the training set?

(d) Some of the results are only reported in text. They should be in tables. Some numbers are just missing. When you report the performance on seen relations, it's really important to provide the numbers for other recent approaches, and not just provide the much less informative ranking. If the paper is short on space, all the tables can be merged into a single table with some separators.

(e) The related work section adds little coming like this at the end, when most of the papers mentioned (if not all) were already discussed at similar level (or deeper) before.

Some more minor issues that the authors should address:

(a) The methods seems to require enumerating and distinguishing domains. It's not specified how this is done in KB used in the paper. This should be made clear.

(b) What is "terms" in Section 2 referring to? Is this a non-standard way to refer to tokens.

(c) For mention detection, why not use B-I-O tagging? This is the most common standard and seems like a perfect fit here. The current choice seems sub-optimal.

(d) In RP, the embeddings are initialized with word2vec, but what about the entity placeholder token? Also, do you use any initialization or embedding sharing between the natural language and the relation tokens?

(e) For AS, the paper mentions using a heuristic based on popularity. Does this really address the problem or maybe works just because of some artifacts in the data? It's OK to use this heuristic, but a 1-2 sentence discussion would help.

(f) The first paragraph in Section 4.1 is confusing with how it sets expectations for what is described below it. For example, the mention of Wikipedia sentences is confusing. It's clarified later, but still. Again, figure and examples would help a lot. The mention of randomly initialized embeddings (next paragraph) is confusing without mentioning training.

(g) Some typos: "... we create a extract of ...", "... users with the same intend many ..."

(h) Why take the median for evaluation? Is it strictly better than mean and stddev?

(i) The use RQx for research questions is not working. The reader just can't remember what each is referring to.

---

> ### Author Response · Authors · 2020-04-14
> **Response to Reviewer 1**
>
> Thank you for your valuable comments. We have revised parts of the paper (including additional experiments and analysis) and marked the changes in the revised version in purple.
>
> 1. The paper provides few examples and no figures...also critical in the experimental results, which provide little insight into the numerical results (e.g., through qualitative analysis).
>
> We have improved the presentation of the approach by using a running example throughout the description of the KGSQA system in Section 3 (“who directed the godfather”).
> Furthermore, we have performed qualitative error analysis to gain further insights into the numerical results in Section 6. Also, we conduct an empirical error analysis by sampling 60 misclassified examples, 10 for each target domain, and investigate what type of error led to the wrong prediction.
>
> 2. There's no evaluation (or even examples) of the generated questions.
>
> We have added evaluation of the generated questions w.r.t. BLEU (Table 5); the findings agree with Table 2: our QG method outperforms the baseline method.
> Additionally, we present success and failure cases of our QG method in Table 4.
>
> 3. During experiments, do you train once and then tune and test for each unseen domain?
>
> We train 6 different models (one per unseen domain) and average the results. In each one of these experiments, the training data consists of the question-facts that appear in the source domains. The gold questions of the corresponding unseen domain are not included in the training step -- only the synthetically generated ones. In detail, we have 82 unique domains. Therefore, our training data consists of 81 domains with gold questions, excluding the one unseen domain. We provide more details in Section 5.
>
> 4. When you report the performance on seen relations, it's really important to provide the numbers for other recent approaches, and not just provide the much less informative ranking.
>
> Since the KGQA performance on seen domains is not the focus of the paper and because of space restrictions we put Table 6 in the appendix. We hope this is satisfactory to the reviewer.
>
> 5. The related work section adds little coming like this at the end, when most of the papers mentioned (if not all) were already discussed at similar level (or deeper) before.
>
> We have revised it to emphasize parts that were not already discussed.
>
>
> Minor
>
> 1. The methods seem to require enumerating and distinguishing domains. It's not specified how this is done in KB used in the paper.
>
> We define domains in the first paragraph of the problem statement. More specifically for Freebase, relations are arranged in a file directory-like hierarchy X/Y/Z (e.g. ``music/album/genre"), where X is the domain.
>
> 2. What is "terms" in Section 2 referring to?
>
> Changed to “tokens”.
>
> 3. For mention detection, why not use B-I-O tagging?
>
> Indeed, B-I-O tagging could also be used. We follow previous work on KGSQA [1,2].
>
> 4. In RP, the embeddings are initialized with word2vec, but what about the entity placeholder token? Also, do you use any initialization or embedding sharing between the natural language and the relation tokens?
>
> The entity placeholder token is randomly initialized. The question and relation tokens share the embedding matrix.
>
> 5. For AS, the paper mentions using a heuristic based on popularity. Does this really address the problem or maybe works just because of some artifacts in the data? It's OK to use this heuristic, but a 1-2 sentence discussion would help.
>
> It is a common heuristic in KGSQA that deals with the fact that there is no explicit linguistic signal to disambiguate the choice [1,2]. In scenarios such as conversational search it is possible to infer this information from the dialogue history.
>
> 6. Typos
>
> Thank you, fixed.
>
> 7. Why take the median for evaluation? Is it strictly better than mean and stddev?
> State-of-the-art works in SimpleQuestions report results for a single model [2-5]. In our work, in order to alleviate the impact of the seed of the random generators, we ran our experiments with a number of different random seeds. However, in order to still be comparable to the rest of the literature, we report the median instead of the mean. Moreover, by reporting the median we are capable of doing statistical significance testing, which requires the predictions of a single model. Lastly, in our experiments the median was not strictly better than the mean.
>
> [1] Mohammed et al. Strong baselines for simple question answering over knowledge graphs with and without neural networks. NAACL 2018
> [2] Petrochuk et al. Simplequestions nearly solved: A new upperbound and baseline approach. EMNLP 2018
> [3] Yu, et al. Improved neural relation detection for knowledge base question answering. ACL 2017
> [4] Gupta et al. Retrieve and re-rank: A simple and effective IR approach to simple question answering over knowledge graphs. FEVER 2018.
> [5] Zhao et al. Simple Question Answering with Subgraph Ranking and Joint-Scoring. NAACL 2019

---

### Official Review · AnonReviewer3 · 2020-03-27
**Simple yet reasonable method for KBQA domain adaptation**

**Rating:** 6
**Confidence:** 3

**Review:**

This paper presents a simple approach for domain adaptation in Knowledge Graph Question Answering. The paper consider the setting where the knowledge graph used to back the QA system contains the necessary facts for a test-time domain, but the training domain didn't cover an examples that required inference over that subdomain. To bridge the gap, the paper proposed a simple procedure for constructing synthetic questions over the relations in the test domain.

Pros:
- The task definition considered is appealing, and identifies another area in which SimpleQuestions is not "solved".
- The approach yields modest but consistent empirical gains across the different domains considered.
- It is relatively simple with few assumptions, making it more likely to generalize.

Cons:
- The novelty of the paper is fairly limited. Synthetic question generation has been well studied in other areas of QA, including training models from synthetic data only. Domain adaption is also well studied; Wu et. al. (cited here) also study adaptation to unseen relations for KBQA (which is inherently closely related to unseen domain adaptation).
- Though not a flaw per se, the generation method is fairly simplistic --- which might work well for something like SimpleQuestions (which hardly require natural language understanding), but not for datasets with richer language structure.
- The empirical gains are small; most of the benefit seems to be coming from the improved RP network, which uses standard components.
- None of the submodules are pre-trained, it would be interesting to see if using a pre-trained encoder such as a large language model (BERT, etc) would help in covering the gap in linguistic variation & understanding across domains.

---

> ### Author Response · Authors · 2020-04-14
> **Response to Reviewer 3**
>
> Thank you for your valuable comments. We have revised parts of the paper (including additional experiments and analysis) and marked the changes in the revised version in purple.
>
> 1. Though not a flaw per se, the generation method is fairly simplistic --- which might work well for something like SimpleQuestions (which hardly require natural language understanding), but not for datasets with richer language structure.
>
> Indeed, questions with complex language structure might be overall harder to tackle. Nevertheless, as stated in [3], even though SimpleQuestions does not require deep language understanding, the task of KGSQA is a task that is yet unsolved. Furthermore, in [1] it is claimed that in practice users tend to ask simple questions with a single entity instead of more complex questions, which makes our task widely applicable in real world scenarios.
>
> Additionally, the pipeline we have at this point can be potentially adapted to work for complex questions. In detail, complex questions are questions that include phenomena such as conjunctions, comparatives, compositions etc. One idea would be to treat complex questions as multiple simple ones, which are combined with some functions. Therefore, we can build on works such as [2] where they learn how to decompose a complex question into simple ones, and also how to combine the respective answer of each simple question in order to give the final answer for the initial complex question.
>
> [1] Iyyer et al. Search-based neural structured learning for sequential question answering. ACL 2017
> [2] Talmor et al. The web as a knowledge-base for answering complex questions. NAACL 2018
> [3] Zhao et al. Simple Question Answering with Subgraph Ranking and Joint-Scoring. NAACL 2019
>
> 2. The empirical gains are small; most of the benefit seems to be coming from the improved RP network, which uses standard components.
>
> Even though the empirical gains are indeed modest, they are statistically significant, while using a simpler RP model than the best baseline model.
>
> 3. None of the submodules are pre-trained, it would be interesting to see if using a pre-trained encoder such as a large language model (BERT, etc) would help in covering the gap in linguistic variation & understanding across domains.
>
> We agree that this could be an interesting experiment, and is something we plan to work on as future work. In the limited time we were given, we tried to investigate this by running a preliminary experiment where we replaced the relation prediction (RP) module--the most important component of the pipeline-- with a BERT-based ranker. The input to BERT is the question and the relation separated by [SEP] (e.g. [CLS] who wrote pulp fiction [SEP] film film written by [SEP]), and we predict a binary label using a linear layer on top of the [CLS] token.
>
> We explored what we could achieve with a BERT architecture in three setups, no training data for the target domain (zero-shot), use our QG method to generate training data for the target domain (synthetic), use gold standard training data from the target domain (gold). The results are as follows:
>
> Setting    Model  Macro-acc.  Micro-acc.
>
> zero-shot  Ours   30.21            29.06
> zero-shot  BERT   47.62            45.29
>
> synthetic  Ours    69.86            70.95
> synthetic  BERT    59.08           59.57
>
> gold          Ours    87.85             88.76
> gold          BERT    81.92            84.90
>
> We observe that BERT generalizes better to the unseen domain when no gold or synthetic data are provided for that domain; an increase of ~17% over our model. We also observe that when synthetic or gold data is provided the BERT performance improves compared to the zero-shot scenario.
> Interestingly, its performance is worse than our model’s (LSTM-based) performance when synthetic or gold data are used. We hypothesize that this is because BERT requires much more data to be further fine-tuned. On the other hand, our model does not generalize that well when no gold or synthetic data are provided but this gives it the chance to get faster biased towards an unseen domain with gold or noisy data. At this point, it is important to highlight again that the augmentation of the training set with synthetic data originating from the target domain has a positive impact on both models (BERT and ours).
>
> To conclude, we would like to underline that our approach mostly focuses on generating synthetic data for the unseen domain, and therefore all the submodules of mention detection, relation prediction, candidate generation, and answer selection can be replaced easily with other models than the ones we present in our work.
>
> Grid search hyperparameters for BERT:
> Learning rate: {1e-6,  1e-5}
> Dropout: {0.5, 0.3, 0.2, 0}
> Output Layer: {[768, 256, 1], [768, 128, 1], [768,1]}
> BERT: {Frozen, trainable}
> # epochs: 1-5
> # negatives: 1-10
>
> We are happy to report the above experiment in the final version of the paper if the reviewer thinks it is necessary.

---

### Official Review · AnonReviewer2 · 2020-03-30
**A data-centric domain adaptation method for simple KBQA that is marginally novel**

**Rating:** 5
**Confidence:** 4

**Review:**

This paper studies the problem of answering "first-order" questions (more on the terminology later) that correspond to a single fact in a knowledge graph (KG) and focuses on the cross-domain setting where no curated training examples are provided for the unseen test domain. The proposed base KGQA model is modified from the state-of-the-art model on SimpleQuestions from (Petrochuk and Zettlemoyer, 2018) but with the relation prediction component changed from a classification model to a ranking model to better handle unseen relations (more on this later). The key contribution is a way of generating synthetic questions for the relations in the unseen domain for data augmentation. The generation model is from (ElSahar et al., 2018) but is augmented with relation-specific keywords mined from Wikipedia via distant supervision. Evaluation on reshuffled SimpleQuestions shows that the proposed method can achieve a reasonable performance on 6 selected test domains of large to moderate scale, and the question generation strategy is better than several baselines.

Strengths
- Overall the paper is well-written and easy to follow
- Cross-domain semantic parsing/question answering is a very important problem because of the broad applicability of the technique.
- The evaluation appears to be well designed and shows some interesting and solid results.

Weaknesses
- Overall the technical contribution appears to be marginal: it's largely a recombination of known techniques for a simpler version of a widely-studied problem - cross-domain semantic parsing.
- The paper rightfully points out the importance of the cross-domain setting. It is, however, a bit surprising to see that the discussion of related work is entirely confined to the works on SimpleQuestions. For a number of clear reasons, building semantic parsing models/training methods that can generalize across domains is a well-recognized demand and has received much attention. It is, for example, a built-in requirement for a number of recent text-to-SQL datasets like Spider. Even just focusing on knowledge graphs/bases, there has been many studies in recent few years. See several early ones listed for references in the end. I'd note that the setting of this paper is sufficiently different from most of the existing studies because it only focuses on questions that correspond to a single fact, but it'd benefit the readers to better position this work in the broader literature.
- The necessity of the proposed modifications to the base KGQA model doesn't seem totally necessary to me. Why not just use the state-of-the-art model from (Petrochuk and Zettlemoyer, 2018) and augment it with the generated questions, or at least use it as a baseline? There might need a few minor adjustments to the base model but it doesn't seem to me it would be substantial. The motivation for a ranking-based relation prediction model is given as "this way we can in principle represent any relation r ∈ R during inference time." However, this doesn't seem to be a very convincing argument. When a new domain is added, in order to apply the proposed data augmentation we would need to re-train the KGQA model. At that point we would have known the relations in the new domain (for the purpose of data augmentation), so why couldn't we train the multi-class classifier again on the augmented data with the new relations added?
- There are a number of places in the proposed method that builds "popularity" as an inductive bias into the method. For example, answer selection always selects the one with the most popular subject entity; in order for the Wikipedia-based distant supervision to work the entity pairs of a relation are required to exist on (and linked to) Wikipedia, which only a fraction of popular entities in Freebase do. Related to that, evaluation is also only conducted on domains that are well-populated in Freebase. This is not desired for cross-domain semantic parsing because: (1) it's more of an artifact of current datasets, and (2) cross-domain semantic parsing is more valuable if it could work for less popular domains (the "long tail"); for popular domains, it's more likely one may be willing to pay the cost of data collection because the incentive is higher.

Minor:
- Personally I don't think "first-order" is the best term for this type of question answering because it's easily confused with first-order logic (the description logics behind semantic web/knowledge graphs is a subset of first-order logic). "Simple" or "single-relational", though still not perfectly precise, may be slightly better if we have to give it a name.


[1] Cross-domain semantic parsing via paraphrasing - EMNLP'17
[2] Decoupling structure and lexicon for zero-shot semantic parsing. EMNLP'18.

---

> ### Author Response · Authors · 2020-04-14
> **Response to Reviewer 2**
>
> Thank you for your valuable comments. We have revised parts of the paper (including additional experiments and analysis) and marked the changes in the revised version in purple.
>
> 1. The paper rightfully points out the importance of the cross-domain setting...but it'd benefit the readers to better position this work in the broader literature.
>
> Thank you for the comment, we have revised the related work section as per your suggestions.
>
> 2. Why not just use the state-of-the-art model from (Petrochuk and Zettlemoyer, 2018) and augment it with the generated questions, or at least use it as a baseline?
>
> The RP module of (Petrochuk and Zettlemoyer, 2018) is used partially as a baseline (see Table 3).  We use a R-BiLSTM for mention detection instead of the BiLSTM-CRF they used since R-BiLSTM achieved better results during preliminary experiments.
>
> 3. The motivation for a ranking-based relation prediction model is given as "this way we can in principle represent any relation $r \in R$ during inference time." However, this doesn't seem to be a very convincing argument. When a new domain is added, in order to apply the proposed data augmentation we would need to re-train the KGQA model. At that point we would have known the relations in the new domain (for the purpose of data augmentation), so why couldn't we train the multi-class classifier again on the augmented data with the new relations added?
>
> Indeed, in order to apply the proposed data augmentation we would need to retrain the KGQA model. However, by modeling the task as ranking we can indeed *in principle* represent any relation $r \in R$ during inference time, albeit the performance without retraining is low (see Table 2, first row). That being said, we find that, in practice, our ranking method outperforms the state-of-the-art method of (Petrochuk and Zettlemoyer, 2018) that is trained as multi-class classification when we retrain both with our synthetically generated questions.
>
> 4. There are a number of places in the proposed method that builds "popularity" as an inductive bias into the method. For example, answer selection always selects the one with the most popular subject entity; in order for the Wikipedia-based distant supervision to work the entity pairs of a relation are required to exist on (and linked to) Wikipedia, which only a fraction of popular entities in Freebase do.
>
> We acknowledge that this is a limitation in the experimental setup of our work. However, since the distant supervision is not limited by using Wikipedia as a corpus, and we could potentially look into other textual sources beyond Wikipedia such as news articles in future work.
>
>  5. Related to that, evaluation is also only conducted on domains that are well-populated in Freebase. This is not desired for cross-domain semantic parsing because: (1) it's more of an artifact of current datasets, and (2) cross-domain semantic parsing is more valuable if it could work for less popular domains (the "long tail"); for popular domains, it's more likely one may be willing to pay the cost of data collection because the incentive is higher.
>
> We would like to emphasize that how well a domain is populated in Freebase is not taken into account in our setup. In fact, in our experiments, the population of a domain is just the number of questions of that domain that appear in the SimpleQuestions dataset. Specifically, the number of questions in training for each domain, and hence the number of triples in the domain considered, are: film (9405), book (4522), location (6353), astronomy (1284), education (937) and fictional universe (836).
>
> We have performed an additional experiment in order to simulate the “long tail” scenario. More specifically, we evaluate the macro-accuracy on RP (the most crucial module in our system) when using a subset of the training data for the target/unseen domain. We test two scenarios : gold (gold standard questions) and synthetic (synthetically generated questions). We obtain results below. One can think of samples representing the number of triples that the domain has. Note that for testing purposes we use the test data as usual (which assumes of course that the domain is populated).
>
> samples  synthetic gold
> 0               30.21       30.21
> 10             36.77       37.14
> 50             41.93       44.56
> 100          50.14        52.72
>
> We observe that the difference between gold and synthetic is marginal in the low data regime, which shows that the synthetic questions generated by our method can be valuable in the long tail scenario.
> We added the complete results in Figure 1a (Section 6) alongside with a description of the experiment.
>
> 6. Personally I don't think "first-order" is the best term for this type of question answering because it's easily confused with first-order logic...
>
> Agreed. We have changed “first-order” to “simple”. Instead of “first-order KGQA”, we now use “Knowledge Graph Simple Question  Answering (KGSQA)”.

---

### Decision · Program_Chairs · 2020-05-01

**Decision:**

Accept

**Comment:**

This paper studies the problem of simple question answering over new, unseen domains during test time. A domain adaption framework and a seq2seq question generation method have been proposed to tackle this problem and demonstrates significant improvements over the previous baselines.

All the reviewers agreed that this paper is well-written and the results are convincing, but the problem is relatively narrow with a focused contribution.  Several reviewers also questioned whether this paper contains enough technical contributions. Some other issues have been already addressed during the discussion phase (long tail relations, presentation issues, and adding more related work).

However, we recommend accepting the paper considering the simplicity and effectiveness of the approach. We think it would lead to more discussion/future work in this direction.